# Repetition In Repetition Out: Towards Understanding Neural Text Degeneration from the Data Perspective

**Huayang Li**$^\heartsuit$    **Tian Lan**$^\spadesuit$    **Zihao Fu**$^\clubsuit$    **Deng Cai**$^\spadesuit$

**Lemao Liu**$^\spadesuit$    **Nigel Collier**$^\clubsuit$    **Taro Watanabe**$^\heartsuit$    **Yixuan Su**$^{\clubsuit,\diamond}$

$^\heartsuit$Nara Institute of Science and Technology    $^\spadesuit$Tencent AI Lab

$^\clubsuit$University of Cambridge    $^\diamond$Cohere

{li.huayang.lh6, taro}@is.naist.jp   lantiangmftby@gmail.com
{jcykcai, redmondliu}@tencent.com {zf268, nhc30, ys484}@cam.ac.uk

## Abstract

There are a number of diverging hypotheses about the neural text degeneration problem, i.e., generating repetitive and dull loops, which makes this problem both interesting and confusing. In this work, we aim to advance our understanding by presenting a straightforward and fundamental explanation from the data perspective. Our preliminary investigation reveals a strong correlation between the degeneration issue and the presence of repetitions in training data. Subsequent experiments also demonstrate that by selectively dropping out the attention to repetitive words in training data, degeneration can be significantly minimized. Furthermore, our empirical analysis illustrates that prior works addressing the degeneration issue from various standpoints, such as the high-inflow words, the likelihood objective, and the self-reinforcement phenomenon, can be interpreted by one simple explanation. That is, penalizing the repetitions in training data is a common and fundamental factor for their effectiveness. Moreover, our experiments reveal that penalizing the repetitions in training data remains critical even when considering larger model sizes and instruction tuning. Our code is available at https://github.com/gmftbyGMFTBY/Rep-Dropout.

## 1 Introduction

The emergence of neural language models (LM) has led to significant achievements in various text generation tasks, such as machine translation [3, 24, 31], summarization [22], and open-ended text generation [17, 20]. However, in open-ended text generation, neural LMs exhibit a strikingly severe degeneration issue, producing unreasonably repetitive texts, particularly when employing maximum *a posteriori* (MAP) decoding algorithms. As illustrated in Fig. 1, even a well-trained LM [20] may suffer from a severe degeneration issue.

There have been numerous attempts to explain the phenomenon of neural text degeneration, with many attributing this problem to flaws in the learning process. Fu et al. [8] claim that high-inflow words increase the probability of generating repetitions. A collection of studies [11, 15, 26, 34] argue that the likelihood objective is the primary factor, because it has the problem of exposure bias and focuses more on next-token prediction rather than sequence generation. Meanwhile, both Chiang and Chen [4] and Xu et al. [36] assert that the self-reinforcement mechanism can make neural LMs fall into the repetitive loops. Intriguingly, despite the divergence in these explanations, all

37th Conference on Neural Information Processing Systems (NeurIPS 2023).

🗣: It was not possible to know which grade of general an officer was by his collar insignia . However , major generals and lieutenant generals wore two rows of nine buttons in groups of three down the of the overcoat , and brigadier generals wore two rows of eight buttons in groups of two .

🤖: It was not possible to know which grade of general an officer would be required to be given to a particular officer . The Army was not aware of the Army's role in the conflict , and the Army was not aware of the Army's role in the conflict . The Army's role in the conflict was limited to the training of

Figure 1: Illustration of repetitions in human and generated text. The human text is from Wikitext-103, and generated text is by greedy search using the GPT-2 model trained on Wikitext-103. The underlined text is the prompt for generation. The blue words and red words indicate the repetitions in human text and machine-generated text, respectively.

corresponding methods proposed have been shown to effectively alleviate the text degeneration issue. This observation leads us to ponder: *could there exist more fundamental factors that can explain the degeneration issue?*

In this study, we strive to provide a straightforward and fundamental interpretation for the degeneration problem from the data perspective. Our preliminary investigation reveals that repetitive words in the training data play a crucial role in the issue. It is also worth noting that repetitions are not necessarily of low quality, because it is a natural and common phenomenon in human writing [1, 12, 28]. Across five datasets with different domains, we observe a strong correlation between repetition rates in training and generated text. This finding encourages us to further assess the impact of repetitive words in training data by randomly dropping out attention to them during training. Employing repetition dropout, we discover that the repetition rate in generated text can be substantially reduced. Lastly, we reconcile many previous hypotheses with our single explanation, asserting that penalizing repetitions in training data is a key factor to the success of alleviating the degeneration issue.

As large language models (LLMs) gain popularity, it appears that the degeneration issue has been somewhat solved. We investigate the impact of various factors associated with LLMs on degeneration, including increasing model size and training models using instruction-tuning data. Our experiments reveal that penalizing repetitions in training data continues to be crucial in the context of LLMs.

Our contributions are threefold:

- We demonstrate that the proportion of repetitive words in training data has a significant influence on the degeneration issue. Inspired by this finding, we also propose a method, namely, *repetition dropout*, to mitigate the degeneration.

- We find that penalizing repetitions in data is a more fundamental factor for reducing repetitions, which also provides a unified explanation for various existing hypotheses, such as the attributions to high-inflow words, likelihood objectives, and self-reinforcement phenomenon.

- We investigate the influence of factors associated with large language models on reducing repetitions, including model scale and instruction tuning.

## 2 Related Work

The degeneration (or repetition) issue in neural language models (LM), particularly in open-ended text generation, has garnered significant attention in recent years. Previous works have proposed disparate interpretations and solutions for this issue.

Many researchers hypothesize that factors within the learning process contributes to the degeneration issue. Fu et al. [8] assert that the high-inflow words in training data may increase the probability of generating repetitions, where the inflow of a word is defined as the probability sum of all its preceding words. Another factor identified is the likelihood objective. Welleck et al. [34] argue that likelihood objective has a discrepancy with the MAP decoding and focuses more on next-token prediction rather than sequence generation. Thus, they proposed an unlikelihood objective to address the two flaws. Based on the same principle, Lin et al. [15] propose to scale the gradient of specified non-novel tokens, i.e., words in the prefix context at each time step, to alleviate the degeneration issue. Su et al. [26] and Jiang et al. [11] pointed out that the degeneration issue may caused by the high similarity between token representations. Thus, they leveraged contrastive learning to learn a more distinct

representation for each token. In addition, both Xu et al. [36] and Chiang and Chen [4] argue that degeneration is caused by the self-reinforcement phenomenon and Xu et al. [36] address this issue by penalizing the repetitions in pseudo repetitive data.

Apart from issues within the learning process, other explanations for degeneration have been proposed. Many researchers contend that decoding methods [7, 10, 16, 20, 37] is the primary factor. Holtzman et al. [10] argue that word probabilities in human-generated text exhibit high variance and randomness, while high-probability text produced by MAP decoding methods tends to be repetitive and dull. This observation explains why sampling-based decoding methods [7, 10, 16, 20] can substantially mitigate the degeneration issue. Riley and Chiang [21] find that tasks with lower constraints, or larger solution spaces, suffer from more severe degeneration issue. The model architecture [8, 32, 34] and size [14] may also contribute, but the two factors have not been quantitatively evaluated.

However, with so many explanations, understanding the primary cause of the degeneration issue becomes increasingly challenging. In our work, we strive to provide a fundamental explanation for the previous hypotheses in the learning process. While our work may not encompass all the existing explanations, we believe it lays a solid foundation for further understanding of the degeneration issue.

## 3  Background

Language model (LM) aims to estimate the probability of a sentence in natural language according to the chain rule of probability:

$$P(\boldsymbol{x}) = P(x_1)P(x_2|x_1)\dots P(x_L|\boldsymbol{x}_{1:L-1})$$
$$= P(x_1)\prod_{i=2}^{L} P(x_i|\boldsymbol{x}_{1:i-1}), \tag{1}$$

where $\boldsymbol{x} = <x_1, \dots, x_L>$ is a sequence of words with length $L$, and $\boldsymbol{x}_{1:i-1}$ is the previous $i-1$ words, i.e., the context, for predicting word $x_i$.

**Model**  Since attention-based LM [20, 31] has became the backbone of many tasks, we will use GPT-2 model [20] as the main architecture for our empirical studies. The core of GPT-2 model[1] is to use the attention mechanism to update the representation of words in $\boldsymbol{x}$:

$$\text{Attention}(\mathbf{Q}, \mathbf{K}, \mathbf{V}) = \text{softmax}\Big(\frac{\mathbf{Q}\mathbf{K}^T + \mathbf{M}}{\beta}\Big)\mathbf{V}, \tag{2}$$

where $\beta$ is a scalar to control the scale of the attention score, $\mathbf{Q}$, $\mathbf{K}$, and $\mathbf{V}$ are the linear transformation of $\boldsymbol{h}_{1:L} \in \mathbb{R}^{L \times d}$ using matrices $\mathbf{W}_q$, $\mathbf{W}_k$, and $\mathbf{W}_v \in \mathbb{R}^{d \times d}$, respectively. $\boldsymbol{h}_{1:L}$ is the current hidden representation of words $\boldsymbol{x}_{1:L}$, and $d$ is the hidden size. $\mathbf{M}$ is masking matrix that makes sure only the information of $\boldsymbol{h}_{1:i}$ is accessible for $\boldsymbol{h}_i$ at each time step $i$. If the word $x_i$ can perceive the information of $x_j$, then $\mathbf{M}[i][j]$ equals to 0, otherwise $-\infty$.

**Training**  Generally, neural LMs are trained by optimizing the likelihood objective:

$$\mathcal{L} = -\sum_{\boldsymbol{x} \in \mathcal{D}} \sum_{i=2}^{L} \log P(x_i|\boldsymbol{x}_{1:i-1}; \theta)$$
$$= -\sum_{\boldsymbol{x} \in \mathcal{D}} \sum_{i=2}^{L} \log \text{softmax}(\phi(\mathbf{H}_{i-1}))[x_i], \tag{3}$$

where $\boldsymbol{h}_{i-1} \in \mathbb{R}^d$ is the hidden vector of $x_{i-1}$ output by the last layer of an neural LM, e.g., the GPT-2 model [20]. The $[x_i]$ is defined as taking the probability regarding to $x_i$ in the distribution got from softmax. The $\phi(\cdot)$ is a linear layer that transforms the $\boldsymbol{h}_{i-1}$ to logits.

---

[1]GPT-2 also incorporates essential sub-layers such as residual connections and layer normalization. While the model employs a multi-head attention mechanism, we have omitted details for brevity. For a comprehensive explanation of these sub-layers, refer to Vaswani et al. [31] and Radford et al. [20].

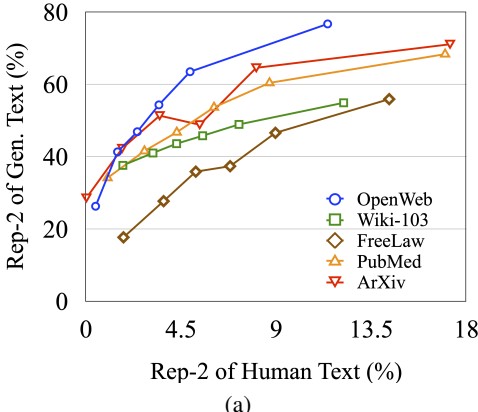 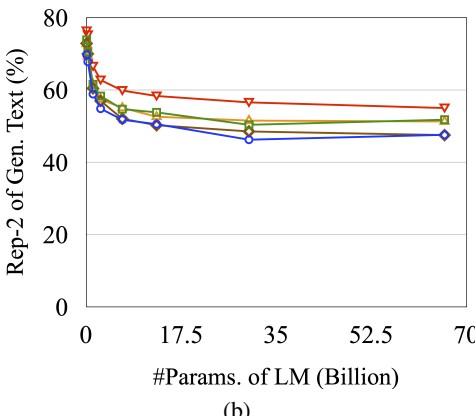

Figure 2: (a) Relationship between rep-2 scores of human and generated text. (b) Relationship between the scale of an LM and the rep-2 score of generated text. Note that results with the same symbol in Fig. 2(a) are from models trained on different shards of the corresponding dataset. The rep-2 score is defined in Eq. (4). We use the GPT-2 and OPT LMs for Fig. 2(a) and 2(b), respectively, and use greedy search as the decoding method.

**Inference**  Since neural LMs are trained to maximize the likelihood objective, one intuitive practice for inference is to use the MAP decoding method, e.g., greedy search or beam search. However, in open-ended text generation, this tactic will cause an extremely severe degeneration issue on vanilla neural LMs trained by likelihood objective [10].

**Evaluation**  The main focus of this empirical study is to investigate the reason for the repetition issue. Therefore, the rep-$n$ is an important evaluation metric in our work, following previous works [8, 25, 27, 34]:

$$\text{rep-}n = 1.0 - \frac{|\text{UniqueNgrams}(\boldsymbol{x}, n)|}{L - n + 1} \quad (4)$$

where $n$ is the length of $n$-gram, and $\text{UniqueNgrams}$ is a function to find all unique $n$-grams in a sentence $\boldsymbol{x}$. For the corpus-level evaluation, we report the averaged rep-$n$ scores of instances in the dataset. To ensure that our results in main experiments are not biased to rep-$n$, we also report the results of rep-$w$ and rep-$r$ in previous works [8, 34]. The rep-$w = \frac{1}{L}\sum_{t=1}^{L} \mathbb{1}\{x_t \in \boldsymbol{x}_{t-w-1:t-1}\}$, which measures the word-level repetition in a prefix window with length $w$. In our experiments, we set $w$ to 16, following Fu et al. [8]. The rep-$r = \frac{1}{L}|\{i|(x_i = x_j \wedge x_{i+1} = x_{j+1}, \exists j \neq i) \vee (x_i = x_k \wedge x_{i-1} = x_{k-1}, \exists k \neq i)\}|$. It is for the portion of repetitive snippets measured by sentence length.

In addition to the measurement of repetition, we also consider the perplexity (PPL) on the real data, which demonstrates the performance in language modeling [2]. Although LMs with lower PPL may not consistently lead to better generation results, it is able to reflect trivial solutions for the degeneration issue, e.g., random or over-fitting models.

## 4  Preliminary Study: Rethinking Neural Text Degeneration from Data Perspective

The propensity of neural LMs with impressive performance to fall into naive repetitive loops is puzzling. Although numerous hypotheses have been proposed, many of them approach this issue from divergent aspects, and some are not intuitive for understanding. According to Ockham's Razor, a simpler explanation is often preferable. Therefore, in our preliminary study, we start by evaluating one elementary factor for most AI systems, the training data. Concretely, repetition is a natural and common phenomenon in human languages for various reasons [1, 12, 28]. It is intriguing to investigate whether there are connections between valid repetitions in natural language and incorrect repetitions in generated language. Moreover, it is important to note that data containing repetitions is not necessarily of low quality, as shown in Fig. 1.

| # | Model | Rep-2 of FT data(%) | Rep-2(%) | Rep-3(%) | Rep-4(%) |
|---|---|---|---|---|---|
| 1 | LLAMA2 W/O FT | – | 47.79 | 41.97 | 38.52 |
| 2 | FT LLAMA2 ON ALPACA | 5.54 | 15.08 | 10.91 | 8.93 |
| 3 | FT LLAMA2 ON ALPACA + WT-103 50K | 9.67 | 41.63 | 35.64 | 32.29 |
| 4 | FT LLAMA2 ON WT-103 50K | 10.31 | 54.10 | 49.77 | 36.80 |

Table 1: Results of LLAMA2-7B on instruction-tuning data. The "FT" means fine-tuning. The column "Rep-2 of FT Data" indicates the rep-2 score of the training data. The rest Rep-$n$ scores are evaluated on the generated text. The ALPACA is the instruction-tuning dataset used in [29], "WT-103 50K" is the instruction-tuning dataset we constructed based on Wikitext-103 (Appx. A.2), and "ALPACA + WT-103 50K" is the mixture of both.

**Setup**  To assess the correlation between repetitions in generated text and those in human text, we propose to train GPT-2 models [20] on data with varying rep-2 scores and then evaluate the rep-2 scores of the text generated by the corresponding models. Specifically, we sorted the training instances in each dataset $\mathcal{D}$ based on their rep-2 scores. We then divided the sorted training data into six shards, each containing an equal number of words but a varying percentage of repetitions, and trained a GPT-2 model on each shard. Notably, we will use the full test set of a dataset $\mathcal{D}$ to evaluate the models trained on different shards of $\mathcal{D}$. More implementation details are in Appx. A.

Our preliminary study investigates five datasets across various domains. Wikitext-103 is a widely used dataset for language modeling [2, 5, 13] and open-ended generation [8, 34]. We adopt the standard split for training, validation, and test sets. The remaining four datasets, OpenWebText2, FreeLaw, PubMed, and ArXiv, are part of the Pile dataset [9]. To ensure consistent analysis, we sample an equivalent number of words as the Wikitext-103 dataset from each of the four Pile datasets. For the validation and test sets, we randomly sample 2,000 sentences from each of the four Pile datasets. The training, validation, and test sets are non-overlapping across all five datasets.

**Findings**  Fig. 2(a) demonstrates a strong correlation between the rep-2 scores of human and generated text on each dataset, indicating that the degeneration issue becomes more severe as the percentage of repetitions in training data increases. However, due to varying data distributions across datasets from different domains, the rep-2 score of generated text may vary even when trained on data with the same rep-2 score. Another intriguing observation is that neural LM will amplify the repetition in data by more than 10 times, similar to the bias amplification found in Zhao et al. [39].

**Investigations Related to Large Language Models**  As LLMs, e.g., ChatGPT [17] and LLAMA[30], gain popularity, it appears that the degeneration issue has been somewhat solved. In this section, we will investigate the impact of various factors associated with LLMs on degeneration, including increasing model size and training models with instruction-tuning data.

Many amazing model abilities emerge when scaling up the model and data size [33]. An interesting question is, *whether the degeneration issue will be solved by simply scaling up*? We use a set of OPT models [38] with different model sizes to investigate this question. As shown in Fig. 2(b), the rep-2 score of generated text sharply drops before increasing the model size to 6.7 billion parameters, indicating that increasing the model size does alleviate the repetition issue to some extent. However, the gains achieved by increasing the model size diminish over time. The OPT-66B model still generates text with high rep-2 score. This observation shows that increasing the model size is not an efficient way to alleviate the degeneration.

Degeneration in certain instruction-tuned LLMs, such as ChatGPT [17], is relatively rare, leading to the hypothesis that the instruction-tuning phase, which trains LLMs on instruction-response pairs, could alleviate this issue. This conjecture implies that the utilization of instruction-tuning data is vital for mitigating degeneration. To explore this, we fine-tune the LLAMA2 model [30] using three instruction-tuning datasets: ALPACA [29], ALPACA + WT-103 50K, and WT-103 50K, as demonstrated in Table 1. The rep-2 scores for these datasets are 5.54, 9.67, and 10.31, respectively. Our experiments reveal that LLAMA2 fine-tuned on ALPACA exhibits less repetitions, while training on WT-103 50K and ALPACA + WT-103 50K still displays significant degeneration. This finding is consistent with our prior observations, where repetition issues strongly correlate with the presence of repetitions in the training data. It suggests that the low repetition rate in instruction-tuning data may contribute to the decreased degeneration.

These investigations reveal that our primary finding, namely, the degeneration issue has a strong correlation with repetitive patterns in training data, remains vital in the context of LLMs. We anticipate that this finding will offer valuable insights for understanding the degeneration issue and contribute to the development of more effective large language models.

# 5 Method

According to our observation in the preliminary study, we hypothesize that learning the patterns of good repetitions in natural language is a crucial factor of the degeneration issue. To better evaluate our hypothesis, we propose a simple method to control the percentage of repetitions that the model can perceive during training, namely, *repetition dropout*. To this end, we can directly measure the impact of the repetition in natural language by training on the full dataset.

## 5.1 Repetition Dropout

Inspired by dropout [23], which prevents the model from over-fitting on specific combinations of parameters, we propose the repetition dropout to avoid the model over-relying on repetitive words in the training phase. Since attention is the core component of Transformer-based LMs to perceive other context words, we apply our repetition dropout to the attention sub-layer[2], which is defined in Eq. (2). More concretely, our method replaces the triangular matrix $\mathbf{M}$ in Eq. (2) by $\mathbf{M}'$:

$$\mathbf{M}' = \mathbf{M} + \mathbf{M}_{rep}, \qquad (5)$$

where $\mathbf{M}_{rep}$ is the masking matrix for repetition dropout. The algorithm for generating the masking vector for each input sentence $\boldsymbol{x}$ is shown in Algorithm 1. In line 5-14 of this algorithm, we define a function find_ngrams to collect all $n$-grams in $\boldsymbol{x}$. The function gen_mask_rep in line 16-26 is to randomly drop out those repeti-tive $n$-grams according a pre-specified repetition

---

**Algorithm 1** Repetition Dropout, Python-like

```
1   # x: List[int], input sequence
2   # p: float, repetition dropout rate
3   # n: int, length of ngram
4
5   def find_ngrams(x, n):
6       # find start and end index of each n-gram
7       ngram_dict = dict()
8       for i in range(n-1, len(x)):
9           ngram = tuple(x[i-n+1:i+1]
10          if ngram in ngram_dict:
11              ngram_dict[ngram].append((i-n+1, i+1))
12          else:
13              ngram_dict[ngram] = [(i-n+1, i+1)]
14      return ngram_dict
15
16  def gen_mask_rep(x, p, n):
17      mask_rep = [0] * len(x)
18      ngram_dict = find_ngrams(x, n)
19      for _, idx in ngram_dict.items():
20          if len(idx) > 1:
21              # mask repetitive n-grams according
22              # to the dropout rate p
23              if random.uniform(0, 1) < p:
24                  for i, j in idx:
25                      mask_rep[i:j] = -inf
26      return mask_rep
```

---

dropout rate $p$, which is between $[0, 1]$. A higher repetition dropout rate indicates that more repetitive $n$-grams are not accessible for the model. We concatenate the masking vectors for sentences in a batch as the $\mathbf{M}_{rep}$. To avoid the model over-fitting on a specific $\mathbf{M}_{rep}$, we generate $\mathbf{M}_{rep}$ for each layer of the LM independently. It should be noted that repetition dropout is only employed in the training phase, analogous to the conventional dropout method [23].

In this method, we compel the model to predict subsequent words without depending on repetitions present in training data. Repetition dropout can be viewed as a means to regulate the quantity of repetitions within the training dataset. If learning on repetitions in training data is a vital aspect, repetition dropout will substantially mitigate the degeneration issue. We want to emphasize that this approach is designed to more effectively assess the significance of repetitions in training data.

# 6 Experiments

**Setup**  To evaluate the effect of repetition dropout, we conduct empirical studies on the Wikitext-103, OpenWebText2, and FreeLaw datasets, which have been introduced in section 4. Compared with section 4, we train the GPT-2 model on the complete datasets rather than splitting them into smaller shards. Please refer to Appx. A for more training details about the GPT-2 model[3].

---

[2]Notice that the repetition dropout is not restricted to attention mechanism. It can also be easily extended to the hidden representation of repetitive words.

[3]https://github.com/huggingface/transformers

| Model | Rep-2(%) | Rep-3(%) | Rep-4(%) | Rep-w (%) | Rep-r (%) | PPL |
|---|---|---|---|---|---|---|
| | | | *Wikitext-103* | | | |
| HI-RE | 41.91 | 33.82 | 28.35 | 38.60 | 66.22 | – |
| SCALEGRAD | 12.49 | 6.85 | 4.44 | 18.31 | 28.98 | 24.72 |
| UL | 36.77 | 28.22 | 22.88 | 39.13 | 61.89 | 21.93 |
| MLE | 47.05 | 38.46 | 32.64 | 46.42 | 72.59 | **21.98** |
| +RAND-DROPOUT | 38.33 | 28.84 | 22.66 | 37.76 | 66.19 | 23.50 |
| +REP-DROPOUT | **9.78** | **4.34** | **2.14** | **22.56** | **25.45** | 28.26 |
| HUMAN | 3.56 | 0.84 | 0.28 | 10.64 | 5.82 | – |
| | | | *FreeLaw* | | | |
| MLE | 51.74 | 46.19 | 42.22 | 39.22 | 73.06 | **16.13** |
| +RAND-DROPOUT | 38.82 | 32.19 | 27.78 | 31.31 | 61.30 | 18.85 |
| +REP-DROPOUT | **10.15** | **5.60** | **3.49** | **17.55** | **23.21** | 20.68 |
| HUMAN | 2.77 | 0.89 | 0.50 | 10.61 | 8.10 | – |
| | | | *OpenWebText2* | | | |
| MLE | 73.96 | 70.61 | 67.91 | 67.28 | 88.27 | **80.37** |
| +RAND-DROPOUT | 63.43 | 57.16 | 52.30 | 58.92 | 82.76 | 91.75 |
| +REP-DROPOUT | **25.24** | **16.14** | **11.10** | **34.73** | **49.80** | 107.02 |
| HUMAN | 4.53 | 1.39 | 0.61 | 12.41 | 13.09 | – |

Table 2: Performances of language models on three datasets. Both RAND-DROPOUT and REP-DROPOUT use a dropout rate 0.6. Rep-$n$ is defined in Eq. (4). The decoding method for all models is greedy search. PPL is the perplexity score on the real test data. Note that we do not report the PPL for HI-RE, because its vocabulary is different from other baselines.

The main baseline method is MLE, which indicates the vanilla GPT-2 model trained by likelihood objective, as defined in Eq. (3). The + REP-DROPOUT is the proposed method, i.e., repetition dropout. We also have several baseline methods. The first one is the + RAND-DROPOUT, which applies dropout on random tokens instead of repetitive tokens. Note that the number of random tokens selected for dropout for this baseline is constrained to the same as the repetitive tokens. We also compare with three previous works that also conducted experiments on Wikitext-103: re-encoding of high-inflow tokens (HI-RE) [8], training with scaled gradient (SCALEGRAD) [15], and unlikelihood objective at token level (UL) [34]. During inference we use greedy search as the decoding method to generate 128 tokens, using the first 32 tokens of each line in the test set as the prompt. More details about those baseline methods are also shown in the Appx. A.

## 6.1 Main Results

As shown in Tab. 2, we evaluate the baseline methods and our method on three datasets. On the Wikitext-103 dataset, both the MLE and MLE + RAND-DROPOUT methods suffer from the severe degeneration issue. More than 33% percent 4-grams in the generated sentences of the two baseline methods are repetitive, which is different from the patterns in natural language. Nevertheless, MLE + REP-DROPOUT can significantly reduce the repetition issue in generated sentences. This observation indicates that repetition in the training data is crucial for the degeneration issue. Compared with methods in previous works in Tab. 2, i.e., HI-RE, SCALE-GRAD, and UL, the REP-DROPOUT also show better performance on alleviating the degener-

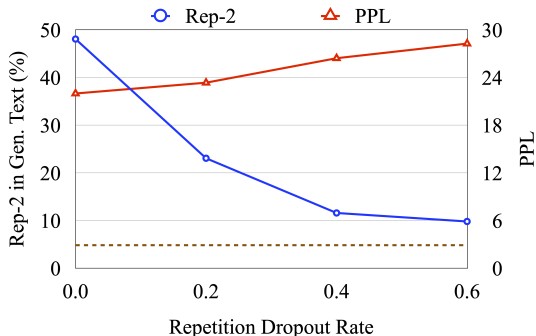

Figure 3: Rep-2 score and perplexity of MLE + REP-DROPOUT on the test set of Wikitext-103. The brown dash line is the human-level rep-2 score.

ation issue. The results on FreeLaw and OpenWebText2 datasets are consistent with those on Wikitext-103. We also conduct quantitative and qualitative experiments to evaluate + REP-DROPOUT on larger LMs, e.g., GPT-XL[20], which are shown in Appx. B and C, respectively.

We also evaluate the effect of MLE + REP-DROPOUT with different dropout rates for the repetitions in Wikitext-103, which can be regarded as controlling the percentage of repetitions in a dataset. As shown in Fig. 3, increasing the dropout rate of our method will reduce the number of repetitions in generated sentences continuously. Moreover, we observe a clear trade-off between the rep-2 score of generated sentences and the perplexity on the real data in test set. This suggests that learning on the repetitions in training data can be beneficial for language modeling. Consistent results are also shown on FreeLaw and OpenWebText2 datasets.

## 6.2 Relation to Previous Hypotheses

Previous research has proposed various hypotheses and approaches to understanding and solving the problem of degeneration in neural text generation. However, we argue that many of these proposals can be explained by a simple explanation. That is, penalizing repetitions in data is a common and fundamental factor for their success.

The core idea of many previous works is to penalize a specific set of data, e.g., all the prefix words $x_{1:t-1}$ at timestep $t$, to alleviate the degeneration. However, as shown in Fig. 4, many of them have implicit connections with the repetitions in data. For example, the set of high-inflow words ($\diamond$) penalized in Fu et al. [8] has a noticeable interaction with that of repetitive words ($\bigcirc$), and the set of prefix words ($\square$) penalized in [34] and [15] is the super-set of the repetitive words ($\bigcirc$). In Xu et al. [36], they directly penalize the repetitions in pseudo repetitive data ($\bigcirc$). In this section, we will show that repetitive words play an important role in previous works.

**High-Inflow Words**  Some researchers [8] attribute the degeneration issue to high-inflow words, whose probability sum of all the potential preceding words is higher than a threshold. Thus, they propose that merging high-inflow word pairs can alleviate the problem (HI-RE). However, we find that 26% of high-inflow word pairs are repetitive in each sentence of Wikitext-103, and merging these pairs can significantly reduce the rep-2 score of real data. Therefore, we argue that merging high-inflow word pairs is actually an alternative way of reducing repetitions in training data.

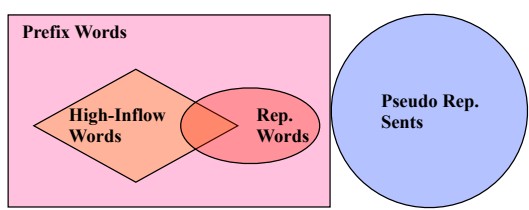

Figure 4: Relationship between the penalized data in previous works. We use $\diamond$, $\square$, and $\bigcirc$ to represent the sets of high-inflow words [8], prefix words [11, 15, 26, 34], and pseudo repetitive sentences [36], respectively. We also demonstrate the set of repetitive words in real data by $\bigcirc$.

We evaluate the argument by controlling a single variable, the type of high-inflow words to be merged, in line 4-6 of Tab. 3. The vanilla HI-RE method (Line 4) merges all the high-inflow words ($\diamond$), which takes 31.1% of the total training words. We find that the method (Line 5), which only merges repetitive high-inflow pairs, i.e., the intersection between high-inflow words and repetitive words ($\diamond \cap \bigcirc$), achieves performance comparable to the original HI-RE method (Line 4). Note that the method in line 5 only merges 8.1% of the total training words, which is much less than the vanilla method. In contrast, the HI-RE method that merges random high-inflow pairs (Line 6), which has the same number as repetitive high-inflow words ($\diamond \cap \bigcirc$), cannot alleviate the degeneration. This suggests that penalizing repetitions in data is critical in the success of Fu et al. [8].

**Likelihood Objective**  Many researchers [11, 15, 26, 34] think that the likelihood objective is the main factor for the degeneration issue. All of those works share the same principle that words in the prefix context ($\square$) cause the degeneration issue. Therefore, Welleck et al. [34] and Lin et al. [15] propose to reduce the probabilities of repetitions appearing in $x_{1:t-1}$ at time step $t$. Su et al. [26] and Jiang et al. [11] leverage the contrastive learning to ensure that the hidden representation at time step $t$ is distinctive to those of $x_{1:t-1}$.

However, the attribution to the likelihood objective merely serves as a superficial explanation, and the core of this issue lies in the fact that the model inevitably learns repetitive behavior when conducting maximum likelihood estimation on repetitive data. As shown in section 6.1, the GPT-2 model with simple repetition dropout, which is also optimized by likelihood objective, achieves an extremely low rep-4 score on generated text. This indicates that likelihood objective might not be the most

| # | Model | Penal. Scope | Word Percent (%) | **Rep**-2(%) | **Rep**-3(%) | **Rep**-4(%) | PPL |
|---|-------|--------------|------------------|--------------|--------------|--------------|-----|
| 1 | MLE | N/A | 0.0 | 47.05 | 38.46 | 32.64 | 21.98 |
| 2 | +Rep-Dropout | Subset of ◯ | 11.4 | **9.78** | **4.34** | **2.14** | 28.26 |
| 3 | DITTO | ◯ | 25.0 | 44.24 | 34.74 | 28.34 | – |
| 4 | Hi-Re | ◇ | 31.1 | **41.91** | 33.82 | 28.35 | – |
| 5 | Hi-Re | ◇ ∩ ◯ | 8.1 | 43.62 | **33.67** | **27.12** | – |
| 6 | Hi-Re | Subset of ◇ | 8.1 | 52.41 | 43.72 | 37.56 | – |
| 7 | ScaleGrad | ▭ | 100.0 | **12.49** | **6.85** | **4.44** | 24.72 |
| 8 | ScaleGrad | ▭ ∩ ◯ | 19.0 | 17.53 | 10.38 | 6.94 | 23.33 |
| 9 | ScaleGrad | Subset of ▭ | 19.0 | 22.97 | 15.22 | 10.94 | 23.18 |

Table 3: Impact of penalizing repetitions in different kinds of data on Wikitext-103. The meanings of the symbols ◯, ◇, ▭ are illustrated in Fig. 4. The "Subset of SHAPE" means a subset randomly sampled from SHAPE. The word percent is $\frac{\text{\#Penalized Words}}{\text{\#Words in Data}}$.

important factor in the degeneration issue. Nevertheless, all methods that penalize tokens in the prefix context (▭) break the reliance on repetitions (◯). Thus, we hypothesize that preventing the model from learning on repetitions is a key factor in their success.

To evaluate the impact of repetitions on these methods, we conduct experiments following the same principle used to analyze high-inflow words. We choose the SCALEGRAD [15] method as our baseline, which penalizes non-novel tokens by scaling the gradient [15]. The non-novel tokens are the entire prefix context $x_{1:t-1}$ (▭) at time step $t$, i.e., $100\%$ of the training words, for the vanilla SCALEGRAD method. We also propose two variants of SCALEGRAD: the first uses the repetitive words within prefix (▭ ∩ ◯) as the non-novel tokens, while the second randomly samples a subset, which has the same number of words as the first one (▭ ∩ ◯), from the prefix data (Subset of ▭). The two methods only penalize $19.0\%$ of the total training words. As shown in Tab. 3, the SCALEGRAD method on repetitive words (Line 8) achieves performance close to the standard SCALEGRAD method (Line 7) in terms of the rep-$n$ metric. In contrast, the SCALEGRAD method on random subset (Line 9) is not effective in alleviating the degeneration issue as the the other two methods. Other works [11, 26, 34] that attribute the degeneration issue to likelihood objective also penalize tokens in prefix as the SCALEGRAD [15], but with different techniques. Thus, we think that penalizing repetitions in data is also a crucial factor for the success of these methods.

**Self-reinforcement Phenomenon**  Both Xu et al. [36] and Chiang and Chen [4] find that degeneration is always accompanied by the self-reinforcement phenomenon, i.e., the probability of a predicted word becomes higher when it is repeated more times. Thus, they hypothesize that degeneration is (partially) caused by self-reinforcement. To mitigate this issue, [36] proposed a data-augmentation method, namely DITTO (Line 3 of Tab. 3), which constructs pseudo data by repeating a training sentence multiple times and penalizes the probabilities of repetitive tokens in the pseudo data (◯).

We argue that, as the degeneration issue, the self-reinforcement is also a by-product when neural LMs learning on repetitive patterns in real data. First, we observe a similar self-reinforcement phenomenon on the repetitive words in real data. For instance, the probabilities of the second appearances of the theme-related words are generally higher than their first appearances, as shown in Fig. 5(b). Second, we find that the model trained by repetition dropout can break the self-reinforcement loop, as shown by the examples in Appx. C. Although the text generated by GPT-2 + REP-DROPOUT on Wikitext-103 may contain a few inappropriate $n$-gram repetitions, it will not fall into the infinite repetition loop that frequently appear in the text generated by the vanilla GPT-2.

### 6.3  Why LMs Learn the Repetition Patterns?

In the last empirical study, we make an attempt to investigate the reasons behind LMs learning repetition patterns, specifically the role these repetitions play in neural LMs. To address this inquiry, we examine 300 randomly selected instances, each featuring repetitive bi-grams within a 256-word sentence. These cases are broadly categorized into three groups, as per Altmann and Köhler [1] and Tannen [28]:

• **Grammar**: Repetitions for grammatical purposes, such as determiners, conjunctions, etc.
• **Theme**: Repetitions closely associated with the subject matter of the text.
• **Limited inventory**: Repetitions resulting from a language's restricted means of expressing a particular concept, leading to high-frequency occurrences, e.g., the phrase "pair of" in English.

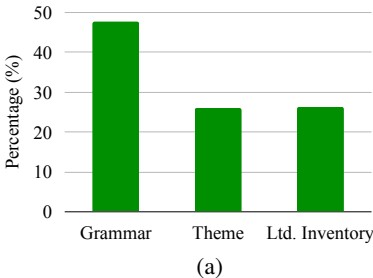 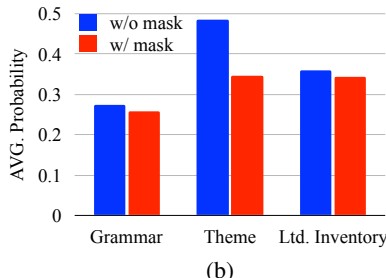

(a)                                               (b)

Figure 5: (a) Percentages of three types of repetitions in Wikitext-103. (b) Probabilities of three types of words with and without masking their repetitions in context using the vanilla GPT-2 model.

We assign each case to the earliest category it satisfies if it meets the criteria for multiple categories. Detailed guidelines for human evaluators to classify repetitive words can be found in Appx. D. As demonstrated in Fig. 5(a), almost 50% of the repetitions in Wikitext-103 fulfill grammatical functions, while both theme-related and inventory-related repetitions constitute around 25% each.

To understand the impact of different types of repetitions on a vanilla neural LM, we measure the probability change of a prediction when masking the information of its repetitions in the context through attention mechanism. For example, given the human repetitions in Fig. 1, we will determine how the probability of the last appearance of "generals wore..." changes when masking its first appearance of "generals wore..." in the context.

As shown in Fig. 5(b), masking the theme-related repetitions will cause a noticeable drop of the prediction probabilities. However, the prediction of grammatical and inventory-related repetitions are not affected by masking their previous appearances in context. This observation indicates that neural LMs spend its effort on optimizing the prediction accuracy of those theme-related words by implicitly repeating words in context.

# 7   Conclusion & Limitations

In this work, we find that the repetition in training data is a fundamental factor for the degeneration (or repetition) problem. First, training data is an integral part of most AI systems, and our preliminary study demonstrates a strong correlation between the repetitions in training data and the degeneration issue. Second, we find that simply dropping out the attention to repetitions in training data can significantly reduce the degeneration issue, which is more effective than other baselines. Finally, we conduct extensive empirical analyses to demonstrate that penalizing repetitions in data is the key success factor for many previous works, such as those attribute degeneration issue to high-inflow words, the likelihood objective, and the self-reinforcement mechanism. Experiments also show that our findings are critical even in the context of large language models. We hope that the viewpoint we provide to understanding the degeneration can inspire more principled research in the future.

Our work also has some limitations. First, despite the excellent performance on reducing repetitions, the repetition dropout method may hurt the perplexity of the language model. Second, we examine our work mostly on standard benchmark of the language generation task, following previous works [8, 10, 26, 34]. It also deserves to extend our work to large-scale data and model.

# 8   Acknowledgement

The authors would like to express their sincere gratitude to the three anonymous reviewers and the meta reviewer for their insightful comments and constructive feedback. Furthermore, this work was partial supported by the JSPS KAKENHI Grant, under the number 23KJ1594.

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

# A  Implementation Details

## A.1  Preliminary Study

The basic GPT-2 model[4] is trained from scratch on each corpus, which has 12 transformer blocks and 12 attention heads with 768 hidden dimensions. The Huggingface transformers [35] and Pytorch toolkit [18] are used to train the GPT-2 model in the distributed manner on A100 GPU server. The hyper-parameters during training are shown in Tab. 4.

| Hyper-parameter | Value |
|---|---|
| Optimization steps | 100K |
| Test interval | 10K |
| Dropout rate | 0.1 |
| Grad clipping | 1.0 |
| Learning rate | $5e^{-5}$ |
| Batch size | 128 |
| Maximum sequence length | 256 |
| Warmup steps | 10K |
| Learning scheduler | Linear decay |
| Random seed | 0 |
| Number of GPUs | 4 |
| Learning objective | Cross-Entropy Loss |

Table 4: The hyper-parameters during GPT-2 training procedure.

### A.1.1  Our Method

Most of the hyper-parameters for our proposed method are the same as that in Tab. 4 for better variable controlling. The specific hyper-parameters for our proposed method are the length of repetitive $n$-gram and its repetition dropout rate $p$, which are set as 2 and 0.6, respectively.

### A.1.2  Baselines

In this subsection, the specific hyper-parameters for three baselines are described, and most of the hyper-parameters are the same as that in Tab. 5.

| Hyper-parameter | Value |
|---|---|
| *Re-encoding of High-inflow Tokens (*HI-RE*)* | |
| Re-encoding $\gamma$ | 0.03 |
| *Scaled Gradient (*SCALEGRAD*)* | |
| Scale grade $\gamma$ | 0.2 |
| *Token-level Unlikelihood Training (*UL*)* | |
| Rank alpha $\alpha$ | 1.0 |

Table 5: The hyper-parameters of three baselines in this paper.

## A.2  Instruction Tuning

To evaluate the effect of instruction tuning, we conduct experiments on three datasets:

1. ALPACA: The instruction-tuning dataset used by Alpaca [29].

2. WT-103 50K: We randomly sample 50k non-title sentences from Wikitext-103 and convert them to the instruction-tuning format, following [29].

3. ALPACA + WT-103 50K: The mixture of both.

---

[4]Model details can be found at `https://huggingface.co/gpt2`

We use the QLoRA [6] to fine-tune the LLAMA2-7B model [30], due to the limited computational resources. The decoding method for generating text is greedy search. We use the test set of Wikitext-103, which is also converted to the instruction-tuning format, to evaluate the performance of different models. Below is the template used for converting Wikitext-103 to the instruction-tuning format:

```
1 {
2       "instruction": "Please continue writing based on the
              following prefix. The text to be continued should be
              relevant, fluent and informative.",
3       "input": PREFIX, # prefix of a sentence
4       "output": COMPLETION # the completion of the prefix
5 }
```

## B  Additional Experiments

In addition to the repetition measurements (Tab. 2), we also evaluate our method on additional metrics, e.g., MAUVE [19]. As shown in Tab. 6, most of the performances of our method and baseline methods are consistent with the observation in Tab. 2.

|  | MLE(%) | HI-RE | SCALEGRAD | UL | REP-DROPOUT |
|---|---|---|---|---|---|
| MAUVE | 49.70 | 35.83 | 52.80 | 50.06 | 52.20 |

Table 6: Evaluation on MAUVE. The settings are the same as those in Tab. 2.

| Model | Rep-2(%) | Rep-3(%) | Rep-4(%) | Rep-w (%) | Rep-r (%) |
|---|---|---|---|---|---|
| MLE | 54.26 | 49.21 | 45.84 | 66.10 | 37.72 |
| + REP-DROPOUT | 11.36 | 5.80 | 3.67 | 24.39 | 18.19 |

Table 7: Experiments on GPT-XL (1.5 Billion parameters)

In addition to GPT-2 model, we also conducted experiments on model with larger size. We directly apply our repetition dropout method to the GPT-XL model, which has 1.5 billion parameters. Since it is difficult to train GPT-XL model[5] from scratch, we fine-tuned it on Wikitext-103 for 3 epochs. Results in Tab. 7 demonstrate that our method can also alleviate the degeneration of LLMs after fine-tuning on larger model.

## C  Qualitative Analysis

In Fig. 6, we show some generated results of the GPT-XL model and GPT-XL+REP-DROPOUT. The training and inference settings of the two models are described in Appx. B. We can observe that the repetitions in generated text were reduced significantly after employing the REP-DROPOUT method. In addition, the GPT-XL trained by MLE easily falls into a sentence-level repetition, because of the self-reinforcement mechanism [36]. Nevertheless, the GPT-XL+REP-DROPOUT does not suffer from this issue.

---

[5] https://huggingface.co/gpt2-xl

**Input Case 1:**

> Traditional Chinese literary criticism emphasized the life of the author when interpreting a work, a practice which Burton Watson attributes to " the close links that traditional Chinese thought posits

**GPT-XL:**

> between the author and the reader ". The author's life was also a central theme in the works of the early modernist poets, such as the poets of the Ming dynasty, who were influenced by the writings of the early modernist poets. The early modernist poets were also influenced by the writings of the early modernist poets, such as the poets of the Song dynasty, who were influenced by the writings of the early modernist poets. The early modernist poets were also influenced by the writings of the early modernist poets, such as the poets of the Ming dynasty, who were influenced by the writings of the early modernist…

**GPT-XL + Rep-Dropout:**

> between the individual and the collective ". The work was also a source of inspiration for the later works of Chinese @-@ American authors such as John W. Campbell, who wrote in his memoirs that he " learned much from [ the ] book ". In the early 20th century, the Chinese scholar and translator Wang Hui ( 1879 – 1940 ) wrote that the book " is the most important work of Chinese literature ". Burton's translation of the text was published in 1891, and was translated into English by the American scholar and translator John W. D. Rockefeller in 1904. It was the first English translation of…

**Input Case 2:**

> In May 1951, he announced he was retiring from professional football to become sports director at a radio station in Buffalo. He was hired as a sportscaster by

**GPT-XL:**

> WLUP @-@ AM in Buffalo, and was promoted to the position of sports director in 1952. He was also a radio announcer for the Buffalo Bills and the Buffalo Braves. In 1953, he was named the radio play @-@ by @-@ play announcer for the Buffalo Bills. He was also a color commentator for the Bills. In 1954, he was named the radio play @-@ by @-@ play announcer for the Buffalo Braves. He was also a color commentator for the Braves. In 1955, he was named the radio play @-@ by @-@ play announcer for the Buffalo Bills. He was…

**GPT-XL + Rep-Dropout:**

> WLUP @-@ FM in Buffalo, New York, in 1952. In 1953, he joined the CBS radio network as a color commentator. His first assignment was to call the NFL championship game between the New York Giants and the Chicago Bears. The Giants won the game, 24 – 0. After the game, he was asked by a reporter if he had ever seen a game like that. " I have, " he replied. " I've seen a lot of football. " He was also a color commentator for the CBS television series The NFL in 1954. During the 1950s, the NFL was the dominant league in America…

Figure 6: Example of generated text. The input text is in bleu, while the repetitions are in red. Both of the two models are fine-tuned on the Wikitext-103 for 3 epochs, as described in section B. We use greedy search for decoding.

# D    Classification of Repetition Words

We categorize repetitions into three groups, as outlined by Altmann and Köhler [1] and Tannen [28]: *grammar*, *theme*, and *limited inventory*. For each sampled instance, we initially determine whether the repetitive $n$-gram falls under the grammar category, meaning any word in the $n$-gram is a determiner, preposition, conjunction, etc. Next, if the repetitive $n$-gram does not belong to the grammar category, we assess whether any words are closely related to the text's subject matter, thereby placing it in the theme category. For instance, "H. gammarus" is considered part of the theme category when repetitively used in an article about Homarus gammarus. The third category encompasses repetitions stemming from a language's limited means of expressing a specific concept, known as limited inventory. Popular phrases such as "pair of" are examples of repetitive $n$-grams in this category.

In cases where multiple repetitive $n$-grams appear within a 256-word sentence, we only take one into account. If a repetitive $n$-gram satisfies the criteria for more than one category, particularly theme and limited inventory, we allocate it to the earliest applicable category.

