# OpenReview forum: "Repetition In Repetition Out: Towards Understanding Neural Text Degeneration from the Data Perspective"
_NeurIPS.cc/2023/Conference — NeurIPS 2023 poster_

### Official Review · Reviewer_Pdys · 2023-06-13

**Soundness:** 3 good
**Presentation:** 3 good
**Contribution:** 3 good
**Rating:** 7
**Confidence:** 4

**Summary:**

This paper studies the cause of neural text degeneration, i.e. language models tend to generate repetitive loops. They design an experiment showing that text degeneration is correlated with the amount of repetitive text in the training data. Motivated by this finding, they propose repetitive dropout, which applies dropout on the attention weights over repetitive context. They experiment with 3 datasets and demonstrate that their proposed method can greatly reduce repetition. They also re-inspect previous hypotheses:

1. high in-flow words: they show that merging repetitive words contributes to a large portion of the effect of merging high in-flow words.
2. the maximum likelihood objective: they show that even when the model is trained with the maximum likelihood objective, as long as repetitions are penalized, the model does not degenerate much. Thus, the maximum likelihood objective is not the cause of neural text degeneration.
3. self-reinforcement of repetition: the argue that
    a. It is also caused by the repetitive words in the training data.
    b. The self-reinforcement loops are broken by their proposed method.

Finally, they categorize repetitions into three categories and study their frequency along with their effect on models’ degeneration behavior. They find that the *theme* category has the greatest effect.


**Strengths:**

1. Most parts of the paper are easy to follow.
2. Neural text degeneration has been discovered since 2019, but its cause is still unclear. Finding the root cause may interest audiences in this field.
3. Despite the hypothesis being simple, as far as I know, it hasn’t been studied. The experiment design in Section 4 is clever.
4. They proposed a simple method that can reduce repetition effectively.
5. They inspected a few previous hypotheses and the results are aligned with their hypothesis (in general).
6. They inspected the effect of three types of repetitions in the training data, which in my opinion is insightful.

In sum, in my opinion, their analyses of the cause of neural text degeneration is comprehensive and the conclusion is convincing.


**Weaknesses:**

In my opinion, because the main message of this paper is clear enough, the following issues may not be very crucial.

1.  The authors only compare their method with one baseline in Table 1. There are other mitigations for text degeneration, e.g. [14, 22, 11] cited in the paper.
2. Section 6.2 is relatively hard to follow.
    1. Line 248, “The core idea of many previous works is to penalize a specific set of data to alleviate the degeneration.” is vague, though I can roughly figure out what it means after reading the following parts.
    2. Having a brief introduction for the definition of high-inflow words will make the paper more self-contained.
    3. For the Likelihood Objective part, the paragraph at line 283 needs a clearer topic sentence. The connection between the experiment design in paragraph at line 288 and the argument in the previous paragraph is not very clear to me either.
3. At Line 312, the author mentioned “we find that the model trained by repetition dropout can break the self-reinforcement loop”. Some evidence should be provided.
4. At Line 223, “we hypothesize that the exposure bias issue is an important factor for the left repetition, because we find that lower-order n -gram2…”, I couldn't understand the explanation. Some details should be provided too.
repetitions generally appear earlier than higher-order repetitions
4. Given the popularity of large language models, the study on GPT-2-sized models may be less useful for the general audience.


**Questions:**

1. If repetition in the training data is the cause of degeneration, why not simply remove those repetitive text from the training data? This seems to me a more straightforward method than applying attention dropout.
2. If the authors promise to address the issues in the weakness section above, I would be happy to increase my score, because I think the main message is clear and interesting enough.
3. Having some analyses on large language models would make this work more impactful, e.g.
    1.  If I understood correctly, the analyses in Section 6.3 can also be done for LLMs.
    2.  I also wonder what would be an effective way to fine-tune an existing (large) model so it doesn't generate repetitive text.


**Limitations:**

Limitations are discussed in Section 7.

---

> ### Author Rebuttal · Authors · 2023-08-09
>
> **Q: The authors only compare their method with one baseline in Table 1.**
>
> *Table 1: Added experiments of ScaleGrad on FreeLaw*
>
> | | Rep-2 | Rep-3 | Rep-4 | Rep-w | Rep-r |
> |-------------|-------|-------|-------|-------|-------|
> | MLE  | 51.74 | 46.19 | 42.22 | 39.22 | 73.06 |
> | ScaleGrad   | 15.82 | 10.28 | 7.52  | 15.97 | 32.40 |
> | Rep-Dropout | 10.15 | 5.60  | 3.49  | 17.55 | 23.21 |
>
> *Table 2: Added experiments of ScaleGrad on OpenWebText2*
>
> || Rep-2 | Rep-3 | Rep-4 | Rep-w | Rep-r |
> |-------------|-------|-------|-------|-------|-------|
> | MLE| 73.96 | 70.61 | 67.91 | 67.28 | 88.27 |
> | ScaleGrad   | 26.61 | 21.26 | 17.98 | 26.08 | 45.21 |
> | Rep-Dropout | 25.24 | 16.14 | 11.10 | 34.73 | 49.80 |
>
> Thank you for your feedback. In our original paper, we conducted experiments on three datasets, as presented in Table 1. For the Wikitext-103 dataset, we compared our method against four baseline approaches: MLE, ScaleGrad, HI-Re, and UL. However, for the FreeLaw and OpenText2 datasets, we only compared our method with the MLE baseline to demonstrate its effectiveness, because in their original works they didn’t evaluate on the FreeLaw and OpenText2 datasets.
>
> As per your suggestion, we have now included additional baselines for the FreeLaw and OpenWebText2 datasets. Due to the time limit of the response period, we have incorporated the performance of the most effective baseline, ScaleGrad, for these two datasets.
>
> The updated results can be found in Tables 1 and 2. Our methods consistently outperform both MLE and ScaleGrad across the majority of Rep-X metrics. The findings on the two datasets align with those observed on the Wikitext-103 dataset. We will report these additional experiments in the revised paper.
>
> **Q: Section 6.2 is relatively hard to follow.**
>
> Thanks for your valuable suggestions. We will follow your comments and elaborate those parts with more details in the revised version.
>
> **Q: At Line 312...**
>
> Thanks for your valuable suggestion! This conclusion was made by inspecting the generated results of MLE and our method. In the revised paper, we promise to attach a case study to support this conclusion.
>
> **Q: At Line 223**
>
> We appreciate your feedback and will clarify this aspect in the revised version. The statement in question aims to explain why our method's Rep-n score did not reach human-level performance. Upon examining the challenging cases encountered by our method, we identified an error accumulation phenomenon, where higher-order repetitions typically occur following lower-order repetitions.
>
> As discussed in previous works, an LM is more prone to fall into repetitive patterns when facing unseen states, which is referred to as the exposure bias issue. Consequently, when our model encounters lower-order repetitions that were not observed in training, it tends to generate more severe degeneration.
>
> **Q: ... Study on GPT-2-sized models may be less useful for the general audience.**
>
> *Table 3: Rep-2 score of text generated by OPT models on five datasets using greedy search.*
>
> | Dataset\Models | opt-125m | opt-350m | opt-1.3b | opt-2.7b | opt-6.7b | opt-13b | opt-30b | opt-66b |
> |----------------|----------|----------|----------|----------|----------|---------|---------|---------|
> | OpenWeb        | 69.66    | 67.74    | 58.80    | 54.75    | 51.68    | 50.46   | 46.17   | 47.52   |
> | Wiki-103       | 73.77    | 70.50    | 61.47    | 58.24    | 54.62    | 53.73   | 50.29   | 51.70   |
> | FreeLaw        | 72.80    | 69.90    | 60.37    | 56.90    | 51.95    | 50.18   | 48.45   | 47.44   |
> | PubMed         | 72.68    | 69.28    | 61.52    | 57.33    | 54.98    | 52.52   | 51.46   | 51.21   |
> | ArXiv          | 76.25    | 75.16    | 66.46    | 62.67    | 59.75    | 58.25   | 56.49   | 54.92   |
>
> We appreciate your valuable suggestion. Indeed, we initially discussed this aspect in an earlier version of our paper. However, we decided to remove it from the final submission to maintain focus and due to the inability to thoroughly investigate all factors within a single paper. We will reintroduce this discussion in the revised paper.
>
> As shown in Table 1, we evaluated the Rep-2 score of OPT models with parameters ranging from 125M to 66B on the five datasets. The results indicate that increasing the model size does help alleviate the repetition issue to some extent. However, the gains from increasing the model size diminish as the size grows. Notably, the OPT-66B model still generates text with high Rep-2 score.
>
> **Q: ... why not simply remove those repetitive text from the training data?...**
>
> Thanks for the interesting question. Actually, our preliminary study in section 4 is based on  a similar idea. However, it is important to note that human text with repetitive n-grams doesn’t indicate low quality. Discarding such data points could have negative effects on the model training. To address this concern, we proposed the repetition dropout method. This approach selectively masks parts of repetitive n-grams, enabling more efficient utilization of the training data.
>
>  **Q: ... the analyses in Section 6.3 can also be done for LLMs.**
>
> Thanks for your suggestion. Yes, the analyses in section 6.3 can also be conducted on LLMs. We will leave the through analysis about LLMs in our future work.
>
> **Q: ... an effective way to fine-tune an existing (large) model ...**
>
> *Table 2: Experiments on GPT-XL (1.5 Billion parameters)*
> || rep-2 | rep-3 | rep-4 | rep-w | rep-r |
> |-------------------|-------|-------|-------|-------|-------|
> | MLE| 54.26 | 49.21 | 45.84 | 66.10 | 37.72 |
> | MLE + Rep-Dropout | 11.36 | 5.80  | 3.67  | 24.39 | 18.19 |
>
> Thanks for this good question. Our method can be directly extended to LLMs. In Table 2, we use our method to fine-tune the GPT-XL model for three epochs. As shown in Table 2, our model can also significantly alleviate the repetition issue of GPT-XL, which further validates the effectiveness of our proposed approach..

---

> > ### Comment · Reviewer_Pdys · 2023-08-10
> >
> > Thank you for your reply. I enjoy reading your solid analyses. I have raised my score.

---

> > > ### Author Response · Authors · 2023-08-14
> > > **Response to Reviewer Pdys**
> > >
> > > Dear Reviewer Pdys,
> > >
> > > We are grateful for your insightful feedback and your recognition of our efforts to address the concerns. Thank you so much for increasing the scores.
> > >
> > > Regards,
> > >
> > > Authors

---

### Official Review · Reviewer_jpnv · 2023-07-06

**Soundness:** 3 good
**Presentation:** 3 good
**Contribution:** 3 good
**Rating:** 6
**Confidence:** 3

**Summary:**

In this paper, the authors demonstrate that repetition in the training data is a major cause of the neural text repetition problem. They first show a strong correlation between the repetition ratio of training data and generated text. Based on the observation, they propose repetition dropout to prohibit the model from learning repetition in data during training. Experimental results show that repetition dropout significantly addresses the repetition problem compared to previous approaches and penalizing repetition in training data is a key factor for reducing the problem. In analysis, the authors demonstrate that their method specifically reduces repetition caused by subject matter rather than grammar or highly frequent phrases.

**Strengths:**

- The proposed method and supporting experiments are well-motivated and the findings are interesting.
- They significantly reduce the neural text repetition problem compared to previous work.

**Weaknesses:**

- To demonstrate that the distribution of generated texts is close to that of human texts, authors could further utilize metrics such as MAUVE [1].
- Experiments on various parameter sizes would be beneficial for further understanding such as "Does increasing model size enhance the robustness to the repetition problem" or "How are the previous and proposed methods effective as the model size increases?"
- A case study on samples generated from the models would provide further insights.

[1] Pillutla, Krishna, et al. "Mauve: Measuring the gap between neural text and human text using divergence frontiers." Advances in Neural Information Processing Systems 34 (2021): 4816-4828.

**Questions:**

- Is there a possible reason for a relatively high repetition ratio in OpenWebText2?

**Limitations:**

As written in the Limitation section, experiments on various scales could be conducted in future work.

---

> ### Author Rebuttal · Authors · 2023-08-09
>
> **Q: metrics such as MAUVE**
> *Table 1: MAUVE scores on Wikitext-103*
>
> |             | Rep-4 ⬆️| MAUVE ⬆️| PPL  ⬇️ |
> |-------------|-------|-------|-------|
> | MLE         | 32.64 | 49.70 | 21.98 |
> | HI-Re       | 28.35 | 35.83 | --    |
> | ScaleGrad   | 5.01  | 52.80 | 39.11 |
> | UL          | 22.88 | 50.06 | 21.93 |
> | Rep-Dropout | 2.14  | 52.20 | 28.26 |
> |             |       |       |       |
>
> Thanks for your kind suggestion. Most results on MAUVE score are consistent with those on rep-n score. As shown in Table 1, our method and ScaleGrad achieve the best performance on Wikitext-103. One exception is the High-Inflow Re-encoding baseline, which achieves the worst performance in terms of MAUVE.
>
> **Q: Does increasing model size enhance the robustness to the repetition problem? How are the previous and proposed methods effective as the model size increases?**
>
> *Table 2: Rep-2 score of text generated by OPT models on five datasets using greedy search.*
> | Dataset\Models | opt-125m | opt-350m | opt-1.3b | opt-2.7b | opt-6.7b | opt-13b | opt-30b | opt-66b |
> |----------------|----------|----------|----------|----------|----------|---------|---------|---------|
> | OpenWeb        | 69.66    | 67.74    | 58.80    | 54.75    | 51.68    | 50.46   | 46.17   | 47.52   |
> | Wiki-103       | 73.77    | 70.50    | 61.47    | 58.24    | 54.62    | 53.73   | 50.29   | 51.70   |
> | FreeLaw        | 72.80    | 69.90    | 60.37    | 56.90    | 51.95    | 50.18   | 48.45   | 47.44   |
> | PubMed         | 72.68    | 69.28    | 61.52    | 57.33    | 54.98    | 52.52   | 51.46   | 51.21   |
> | ArXiv          | 76.25    | 75.16    | 66.46    | 62.67    | 59.75    | 58.25   | 56.49   | 54.92   |
> |                |          |          |          |          |          |         |         |         |
>
> Thank you for your valuable suggestion. We did discuss this aspect in an earlier version of our paper but decided to remove it from the final submission, as we felt it might detract from the paper's focus. As demonstrated in Table 1, we evaluated the rep-2 score of OPT models with parameters ranging from 125 Million to 66 Billion on the five datasets used in our final paper. Our results show that increasing the model size does alleviate the repetition issue to some extent. However, the gains achieved by increasing the model size diminish over time, and the OPT-66B model continues to generate text with an extremely high rep-2 score.
>
> *Table 3: Experiments on GPT-XL (1.5 Billion parameters)*
>
> |                   | rep-2 | rep-3 | rep-4 | rep-w | rep-r |
> |-------------------|-------|-------|-------|-------|-------|
> | MLE               | 54.26 | 49.21 | 45.84 | 66.10 | 37.72 |
> | MLE + Rep-Dropout | 11.36 | 5.80  | 3.67  | 24.39 | 18.19 |
>
> Because of the time limit of the rebuttal period, we only evaluated our method and one baseline method on GPT-XL, which has 1.5 billion parameters. We fine-tuned the GPT-XL using MLE and our method for 3 epochs. As shown in Table 3, we find that fine-tuning a larger model with our method can also significantly alleviate the repetition issue.
>
> **Q: A case study on samples generated from the models would provide further insights.**
>
> Thanks for the useful suggestion. In the revised version, we will add a case study section to show the characteristics of our method and the baseline method.
>
> **Q: Is there a possible reason for a relatively high repetition ratio in OpenWebText2?**
>
> Thanks for your good question. The data in OpenWebText2 comes from Reddit where each thread is discussing a particular topic with diversity in text style and many people are involved. In the section 6.3, we find that LMs would spend more effort on learning the repetition of theme-related n-grams to reduce the model’s PPL. Consequently, we hypothesize that the Reddit data may encourage the repetition behavior in the learning process. That’s also why we conduct experiments on multiple datasets in section 4.

---

> > ### Comment · Reviewer_jpnv · 2023-08-17
> >
> > Thank you for the experimental results which have addressed my curiosity. I could notice the repetition problem does not resolve by simply scaling the language model and Rep-Dropout is effective for the larger models. I think further studies on scaling and decoding strategies would also be meaningful.
> >
> > I raised the score after reading the rebuttal.

---

> > > ### Author Response · Authors · 2023-08-19
> > >
> > > Thanks for your constructive feedback and raising the overall score! We agree that it is meaningful to investigate the scaling and decoding strategies, and we also planned to compare those different factors in a unified framework.  We will leave the detailed investigations to our future work.

---

### Official Review · Reviewer_6VM2 · 2023-07-07

**Soundness:** 3 good
**Presentation:** 3 good
**Contribution:** 3 good
**Rating:** 6
**Confidence:** 4

**Summary:**

The paper explores the issue of degeneration in text generation, which refers to the generation of repetitive words and dull loops by neural language models. The authors focus on the impact of repetition in the training data and propose a method to address this issue. Specifically, they suggest dropping out repetitive words during the training data pre-processing stage. The main conclusion drawn from their investigation is that penalizing repetitive n-grams in the training data is crucial for the effectiveness of existing methods in preventing degeneration. By highlighting the importance of addressing repetition in the training data, the paper contributes to understanding the factors influencing degeneration in text generation.

**Strengths:**

The paper is generally well-written and is easy to follow. The problem of generating repetitive words and dull loops has been widely observed and extensively discussed in the field, which makes this problem important and worth exploring. The conclusion drawn by the authors, that training data plays a pivotal role in the occurrence of degeneration, is logical and well-supported.

**Weaknesses:**

The paper primarily focuses on data pre-processing rather than novel training techniques. This limited scope may diminish its technical novelty. Many researchers working on text generation may have already explored similar data pre-processing ideas, making the proposed method less impactful or original. Also, the conclusion of the paper lacks novel insights and seems to affirm an intuitive understanding. While it is reasonable to assume that data quality has a direct impact on degeneration, this insight does not offer any new or surprising findings. As a result, the conclusion may be considered weak in terms of offering novel contributions or expanding the current understanding of the issue.

Explore additional factors that could contribute to degeneration in text generation, such as the number of training data and model size. Investigating these factors may provide deeper insights into the phenomenon and allow for a more comprehensive understanding of degeneration. This expansion could strengthen the paper and enhance its impact within the field.

For me, it would be more interesting to study other factors that impact degeneration, such as the number of training data, the model size and the model architecture. Investigating these factors may provide deeper insights into the phenomenon and allow for a more comprehensive understanding of degeneration. With more training data and more parameters, large language models can capture semantics and thus prevent degeneration. For example, degeneration has been greatly resolved for ChatGPT due to its expressiveness and more general understanding of texts,  In my opinion, tackling degeneration in data pre-processing is not very meaningful because the root cause is still the model lacks of capability to understand and generate reasonable texts.

**Questions:**

Does dropping some repetitive n-grams sometimes distort the semantics of the document?

**Limitations:**

There are other potential factors that can impact the degeneration. The paper should have a more comprehensive discussion about all the factors.

---

> ### Author Rebuttal · Authors · 2023-08-09
>
> **Q: While it is reasonable to assume that data quality has a direct impact on degeneration, this finding is not surprising**
>
> We'd like to emphasize that the goal of our work is to investigate the relationship between repetition in training data and degeneration. It is important to note that data containing repetitions is not necessarily of low quality. In fact, the repetition of certain words in natural language is sometimes necessary [1,12,23] (cited in paper). The problem is that LMs will learn and amplify the repetition in training data, which has not been studied in previous works. In addition, we give a unified interpretation about the reason of success of many previous works from the data perspective.
>
> In recent years, data-centric AI has garnered the attention of numerous researchers, as many model behaviors may be linked to patterns in the data. By establishing a clear connection between repetitions in training data and the degeneration issue, we believe that researchers can gain a better understanding of degeneration and develop more targeted and effective methods.
>
>
> **Q: Explore additional factors**
>
> Table 1: Rep-2 score of text generated by OPT models on five datasets using greedy search.
>
> | Dataset\Models | opt-125m | opt-350m | opt-1.3b | opt-2.7b | opt-6.7b | opt-13b | opt-30b | opt-66b |
> |----------------|----------|----------|----------|----------|----------|---------|---------|---------|
> | OpenWeb        | 69.66    | 67.74    | 58.80    | 54.75    | 51.68    | 50.46   | 46.17   | 47.52   |
> | Wiki-103       | 73.77    | 70.50    | 61.47    | 58.24    | 54.62    | 53.73   | 50.29   | 51.70   |
> | FreeLaw        | 72.80    | 69.90    | 60.37    | 56.90    | 51.95    | 50.18   | 48.45   | 47.44   |
> | PubMed         | 72.68    | 69.28    | 61.52    | 57.33    | 54.98    | 52.52   | 51.46   | 51.21   |
> | ArXiv          | 76.25    | 75.16    | 66.46    | 62.67    | 59.75    | 58.25   | 56.49   | 54.92   |
>
> Thank you for your valuable suggestion. In the early version of our paper, we did evaluate factors beyond the data, such as the impact of model architecture and model size.
>
> We obtained some interesting findings. For instance, in Table 1, we assessed the rep-2 score of OPT models with parameters ranging from 125M to 66B on the five datasets used in our final paper. Our results show that increasing the model size does alleviate the repetition issue to some extent. However, the gains achieved by increasing the model size diminish over time. The OPT-66B model still generates text with high rep-2 score. We also evaluated the impact of various model architectures, such as enc-dec Transformer models, dec-only Transformer models, LSTM models, etc. All models trained by MLE exhibit severe repetition issues, with no clear indication of which architecture suffers more from this problem.
>
> Nevertheless, we decided to remove those sections from final submission, which was a difficult decision. The reason behind this is that it is challenging to thoroughly analyze and explain all these factors within a 9-page paper. As a result, we chose to focus on one critical factor, the repetition in training data.
>
> We can reintroduce the discussion about other factors in the appendix of the revised paper.
>
> **Q: The paper primarily focuses on data pre-processing rather than novel training techniques.**
>
> We appreciate your concern and would like to clarify that while our analysis paper includes experiments with data pre-processing to clearly demonstrate the research problem, our method to alleviate degeneration is a learning algorithm, not a data pre-processing method. Specifically, inspired by dropout, we propose an attention-based repetition dropout method to encourage the model to make predictions without relying on repetitive n-grams in the context. As demonstrated in our paper, the LM trained using our method achieves exceptionally low rep-n scores at a much lower cost than scaling up the model and data size.
>
> **Q: With more training data and more parameters, LLMs can prevent degeneration, e.g., ChatGPT...**
>
> We agree that increasing the number of data and model size is likely to improve the performance. However, as shown in Table 1, OPT-60B still suffers from severe degeneration and increasing the size can only alleviate the degeneration issue to a certain extent. E.g., the rep-2 improves from 70 to 47 when the model size increases from 125M to 60B. This fact shows that the degeneration issue can not be mainly attributed to the small scale of model size and training dataset size. Moreover, in our preliminary experiments, lots of LLMs, including GPT-3 (e.g., text-davinci-002, 175B parameters), continue to suffers from the degeneration issue. Fortunately, simply dropping out the attentions relating to repetitions in training data can effectively reduce the repetition to an extremely level even with relatively small scale of model size and training data. Therefore, we think the repetitions in training data is a crucial factor for the degeneration issue.
>
> As you pointed out, we also observed that ChatGPT exhibits less degeneration. However, ChatGPT is a product-level system got from a long development pipeline. To the best of our knowledge, there is no convincing literature within our community explaining how ChatGPT achieved this. In fact, this is a research topic that we are currently exploring.
>
> We will add this discussion to the revised paper to clarify our motivation.
>
> **Q: Does dropping some repetitive n-grams sometimes distort the semantics of the document?**
>
> Yes, it is possible. This is a common potential issue of many previous works, e.g., BERT, which also masks part of the text. That’s why we only apply the repetition dropout in the training time. In other words, the word at each time step can access all the prefix words during the inference time. When conducting human evaluation on the generated results, we also find that the text quality of our method is much higher than the MLE method.

---

> > ### Comment · Reviewer_6VM2 · 2023-08-17
> > **Raise my rating to 6**
> >
> > Thanks for the rebuttal, which addressed my primary concerns. The observation that scaling up a model size can somewhat mitigate repetition while still presenting a significant challenge for larger models is interesting. I also notice that the models discussed in the rebuttal are not instruction-tuned models. I'm curious if there's a specific rationale behind this choice.  Although I recognize ChatGPT is not comparable due to its limited accessibility, I believe it could be insightful to include a comparison of instruction-tuned models (i.e. alpaca/vicuna/Llama2) and the model without instruction-tuning (i.e. LLama). Also, it would be interesting to analyze how rep scores connect to the downstream generative task performance metrics like ROUGE scores for summarization.

---

> > > ### Author Response · Authors · 2023-08-19
> > >
> > > Many thanks for your kind reply to our previous response and raising the scores! We really enjoy the discussion with you, and we are happy that our response addressed your primary concerns,
> > >
> > > **Q: I also notice that the models discussed in the rebuttal are not instruction-tuned models. I'm curious if there's a specific rationale behind this choice.**
> > >
> > > The main reason of this choice is to make a fair comparison with previous works, because most previous works only focus on the standard language model.
> > >
> > > **Q: I believe it could be insightful to include a comparison of instruction-tuned models (i.e. alpaca/vicuna/Llama2) and the model without instruction-tuning (i.e. LLama).**
> > >
> > >
> > > Table 1: Results of Llama2-7B on instruction-following data. The column "Rep-2 of FT Data" indicates the rep-2 score of the training data used for fine-tuning. The rest Rep-2, Rep-3, and Rep-4 scores are evaluated on the generated text by different methods. The "FT" means fine-tuning.
> > >
> > > | No. | Methods                       |   | Rep-2 of FT Data |   | Rep-2 | Rep-3 | Rep-4 |
> > > |-----|-------------------------------|---|------------------|---|-------|-------|-------|
> > > | 1   | Llama2 w/o FT                 |   | --               |   | 47.79 | 41.97 | 38.52 |
> > > | 2   | FT  Llama2  on Alpaca              |   | 5.54             |   | 15.08 | 10.91 | 8.93  |
> > > | 3   |  FT  Llama2  on Alpaca + WT-103 50K |   | 9.67             |   | 41.63 | 35.64 | 32.29 |
> > > | 4   | FT  Llama2  on WT-103              |   | 10.31            |   | 54.10 | 49.77 | 36.80 |
> > >
> > >
> > >
> > >
> > > Thanks for this valuable suggestion. We agree that analyzing the effect of instruction-tuning on degeneration is meaningful. We conduct experiments on three datasets:
> > >
> > > 1. **Alpaca**: The instruction-tuning dataset used by Alpaca [1].
> > > 2. **WT-103 50K**: We randomly sample 50k sentences from Wikitext-103 and convert them to the instruction-following data. More details are at the end of this response.
> > > 3. **Alpaca + WT-103 50K**: The mixture of Alpaca and WT-103 50K
> > >
> > >
> > > As shown in Table 1, the “Llama2 w/o FT” (Line 1) indicates the LLM without instruction-tuning, and “FT Llama2 on Alpaca” (Line 2) means the Llama2 with instruction-tuning. We can find that the instruction-tuning process does alleviate the degeneration issue.
> > >
> > > [1]: https://crfm.stanford.edu/2023/03/13/alpaca.html
> > >
> > > However, we hypothesize that the alleviation of degeneration is caused by that the training data of **Alpaca** has less repetitions. As shown in Table 1, the rep-2 scores of the **Alpaca**, **Alpaca + WT-103 50K**, and **WT-103 50K** datasets are 5.54, 9.67, and 10.31, respectively. We can find that the degeneration issue becomes severer if we fine-tune the model on instruction-following data with higher repetition rate (Line 2-4 in Table 1). This observation further demonstrates that the degeneration issue has a high correlation with the repetitions in training data during the instruction-tuning process, which is consistent with the finding in our paper.
> > >
> > > Implementation details of our experiments:
> > >
> > > 1. **Fine-tuning strategy**: we use the QLoRA to fine-tune the Llama2-7B model, due to the limited computational resources.
> > > 2. **Decoding strategy**: greedy search
> > > 3. **Test Data**: The test set of Wikitext-103 in the instruction-following format.
> > > 4. **Data pre-processing**: To ensure a fair comparison, we convert the wikitext-103 dataset to a instruction-following dataset by using the following template:
> > >
> > > ```
> > > {
> > >         "instruction": "Please continue writing based on the following prefix. The text to be continued should be relevant, fluent and informative.",
> > >         "input": PREFIX, # prefix of a sentence
> > >         "output": COMPLETION # the completion of the prefix
> > > }
> > > ```
> > >
> > > **Q: Also, it would be interesting to analyze how rep scores connect to the downstream generative task performance metrics like ROUGE scores for summarization.**
> > >
> > > Thanks for your suggestion. The downstream tasks, e.g., summarization, may also suffer from the degeneration issue. We agree it is interesting to evaluate our methods and findings on those tasks. We will leave the further investigation to future work.

---

### Official Review · Reviewer_6H9c · 2023-07-09

**Soundness:** 4 excellent
**Presentation:** 3 good
**Contribution:** 3 good
**Rating:** 7
**Confidence:** 4

**Summary:**

This paper explains the repetition in model generated text from a data standpoint, pointing out that there is a strong correlation between the degeneration issue and the presence of repetitions in training data. The authors find out that penalizing repetitions in data can alleviate degeneration, and propose a method, repetition dropout, that is to apply dropout in transformers’ sublayers during training time. The proposed method achieve significant improvement in terms of REP-n scores.

**Strengths:**

- This is a very well written paper! The preliminary study does a great job in introducing and clearly motivating the problem.
- The proposed repetition dropout is also a very simple technique, yet still improve the performance in REP-n stats.
- The comparison between different methods and objectives in section 6 is very interesting and thorough.

**Weaknesses:**

Currently it seems that the repetition dropout mask is applied on each instance, but what about repeated text in different instance? How should we apply it in current LM training paradigm?

**Questions:**

NA

---

> ### Author Rebuttal · Authors · 2023-08-09
>
> **Q: Currently it seems that the repetition dropout mask is applied on each instance, but what about repeated text in different instance?**
>
> Thanks for the question. To ensure that we understand your query correctly, we would like to confirm that by "repeated text" you are referring to repetitive n-grams, and by "instance" you mean a sentence. If our interpretation is accurate, in our work, the decision to drop an n-gram through the attention mechanism is entirely dependent on its context, i.e., a sentence with 256 words. In other words, if an n-gram appears only once in the context, we will not drop it. However, if it occurs multiple times within the context, there is a possibility that we will mask them according to the dropout rate. Please let us know if we have misunderstood your question or if you require further clarification!
>
> **Q: How should we apply it in current LM training paradigm?**
>
> Table 1: Experiments on GPT-XL (1.5 Billion parameters)
>
>
> |                   | rep-2 | rep-3 | rep-4 | rep-w | rep-r |
> |-------------------|-------|-------|-------|-------|-------|
> | MLE               | 54.26 | 49.21 | 45.84 | 66.10 | 37.72 |
> | MLE + Rep-Dropout | 11.36 | 5.80  | 3.67  | 24.39 | 18.19 |
>
>
>
> Thanks for this good question. The only difference between current LM training paradigm and ours lies in the repetition dropout mask, which could be easily pre-computed before pretaining. Thus, our proposed repetition dropout technique is compatible with most of LM pertaining paradigm. For example, it is easy for us to extend our proposed repetition dropout technique to the larger language model. In Table 1, we directly apply our repetition dropout method to the GPT-XL model, which has 1.5 billion parameters. Since it is difficult to trained GPT-XL model from scratch, we fine-tuned it on Wikitext-103 for 3 epochs. Results in Table 1 demonstrate that our method can also alleviate the degeneration of LLMs after fine-tuning.

---

> > ### Comment · Reviewer_6H9c · 2023-08-15
> > **Thank you for your reply**
> >
> > Hi,
> >
> > Thank you for your reply!
> >
> > >confirm that by "repeated text" you are referring to repetitive n-grams, and by "instance" you mean a sentence
> > Yes, your interpretation is correct.
> >
> > I am satisfied of the answers to my question, and my rating will stay the same.

---

> > > ### Author Response · Authors · 2023-08-19
> > >
> > > Thanks for your kind confirmation, and we are glad that the previous response addressed your concern.

---

### Decision · Program_Chairs · 2023-09-21

**Decision:**

Accept (poster)

**Comment:**

All reviewers have agreed that the problem this paper focuses on is important and well-motivated, the paper is well-written, the approach is simple but effective, and the experiments are comprehensive.

After discussion, all reviewers voted to accept this paper, and I believe that this paper can provide some insights about LLM control.
The camera-ready version can be better by including more experiments and explanations shown during the discussion so that the readers can better understand the paper.